# Bibliometric Review of Participatory Budgeting: Current Status and Future Research Agenda

Miloš Milosavljević *[ID], Željko Spasenić [ID] and Jovan Krivokapić

Faculty of Organizational Sciences, University of Belgrade, 11010 Belgrade, Serbia;
zeljko.spasenic@fon.bg.ac.rs (Ž.S.); jovan.krivokapic@fon.bg.ac.rs (J.K.)
* Correspondence: milos.milosavljevic@fon.bg.ac.rs; Tel.: +381-11-3950875

**Abstract:** Participatory budgeting has been advocated as an advanced tool of civic participation and a travelling innovation for more than three decades. This paper provides a bibliometric review of the concurrent body of knowledge on participatory budgeting (PB), explaining how this democratic innovation 'travelled' through time and over different scientific fields. This study was based on a dataset of 396 papers on PB published from 1989 to January 2023. The study finds that research in PB has reached its peak of scholarly attention in pre-COVID-19 pandemic years. The study also finds that the research on PB has migrated from the field of political science to other fields, such as economics, management science, law, urban planning, environmental science, and technology.

**Keywords:** public finance management; participatory budgeting; bibliometric review; literature review; governance innovation

## 1. Introduction

In the dynamically changing world of today, not many people would use innovation as the first adjective to describe the World Wide Web (www) for instance. Nonetheless, for the www's peer—participatory budgeting, we can still hear scholarly voices claiming that it is an innovation. Or at least it has been treated as a travelling innovation among scholars, international organizations, policymakers, and other practitioners (Lehtonen 2022).

The concept of PB was first introduced in the Brazilian city of Porto Alegre in the late 1980s (Manes-Rossi et al. 2021). The concept has been evolving and diffusing globally ever since. On the one hand, in some countries, the best practice cases were created paving the way for deliberate democracy. On the other hand, some countries and regions have only witnessed trivial pursuits for the real implementation of this political, societal, and economic novelty (De Vries et al. 2021).

In a recently published book, Wampler and Goldfrank (2022) analyzed PB dynamics in the country of its origin—Brazil. They infer that 'by 2020, PB was one of the world's most widely adopted participatory programs, but it has largely been abandoned in Brazil, the country where it all began'. Another study finds that PB might be a continual and sustainable concept only if developed by the standards of the local community, and not enforced or 'highly recommended' by international organizations or any other neo-liberal external actors (Milosavljević et al. 2020). Having this in mind, one might put an interrogative to the previously defined bright future of PB in both theory and practice.

Although PB has been circulating in scholarly literature for some time, the concept is still vague and amorphous. Participatory budgeting is a multifaceted phenomenon and has been researched accordingly. Traditional themes that covered PB have tried to explain the different aspects of PB—design (Moir and Leyshon 2013; Gilman and Wampler 2019; Mattei et al. 2022), processes (Cabannes 2004), logics (Bartocci et al. 2019), sustainability (Murray Svidroňová et al. 2023b), or barriers (Trtovac Šabović et al. 2021). Alongside the traditional topics, a scholarly body of knowledge on novel themes has been blooming in the last few decades (Bartocci et al. 2022).

Empirical studies still dominate the spectrum of PB research (at least for the sample of papers used in this study bearing in mind that some sub-fields of PB research, such as computational social choice work on PB, tend to work on axiomatic mathematical proofs and simulation models). These studies are usually underpinning PB in practice from a political, good governance, or technocratic point of view (Cabannes and Lipietz 2017). Additionally, to obtain a profound understanding of PB research flow, a myriad of literature reviews has already been conducted in the field of PB. One of the most influential was conducted by Sintomer et al. (2008) who classified all PBs in four distinct categories. These categories have been widely accepted among scholars and practitioners. Other reviews were focused on the experiences of a single country, such as South Korea (Cho et al. 2020) or Germany (Zepic et al. 2017). Implying the different logics, some reviews focus on subtopics within the PB realm of research. In a recent study done by Bartocci et al. (2022), a systematic literature review was conducted with 139 papers focused on PB aimed at investigating the PB journey. Another near-bibliometric analysis was conducted by Nugra and Mera (2018) with an idea to provide synthetic evidence on the process and success factors of PB. Finally, Pereira and Figueira (2020) conducted systematic research with a large dataset aimed at detecting rationales and barriers to citizen participation in PB. The latter three studies use a large number of publications to produce systematic and interpretative reviews. As such, they share some similarities to the standard bibliometric review. Nonetheless, none of the actual bibliometric analyses have been adopted to investigate PB research so far. This creates a lacuna in the present body of knowledge worthy enough for further investigations.

Having in mind that previous studies have already systematically observed PB as a scholarly field of research, this study extends the concurrent body of knowledge in several directions:

i.　Vertically—we incorporated new papers in the analysis;
ii.　Horizontally—we extended our analysis from viewing only the number of publications by year or most cited papers to more sophisticated bibliometric analyses, such as spatial distribution of papers, cooperative teams, journal and author productivity, and interesting research subtopics (Donthu et al. 2021).

Bibliometric analysis is a quantitative technique used to reflect on the current status and main trends in different fields of research (Tao et al. 2020). It provides many useful outputs, such as total publications, citations, and collaboration among institutions and researchers (Donthu et al. 2021). This technique has been used in both financial (Spasenić et al. 2022; Garg et al. 2023) and public administrative studies (Ni et al. 2017).

In this study, we use a science mapping technique to review publications on participatory budgeting retrieved from the Web of Science Core Collection database. The aim of this study is to quantitatively analyze the global research output and provide some future directions for the PB research domain. The specific goals of our study are to answer to research questions listed below:

RQ1. How propulsive are scholarly publications in the PB field?
RQ2. Is PB as traveling innovation evenly distributed in spatial and cooperative terms?
RQ3. Which are the most productive journals and authors in the field of PB?
RQ4. Which sub-topics dominate the concurrent body of knowledge on PB?

The remainder of this study is organized as follows. Section 2 explains the analytical framework for bibliometric study. Section 3 presents the results. Section 4 contextualizes the findings and provides implications for the main stakeholders. Section 5 is reserved for the conclusions, limitations, and further recommendations.

## 2. Methodology

To understand the evolution of academic contributions to the topic of participatory budgeting, we employed the combined research approach of Ropret and Aristovnik (2019), Agrifoglio et al. (2020), and Spasenić et al. (2022). The first phase refers to the identification of all relevant papers within the Web of Science database (WoS). WoS was chosen as the focal database since it is one the most comprehensive and reliable sources of information for bibliometric studies in the various research areas (Singh et al. 2021). Also, recent bibliometric studies and review papers in public administration use WoS as the primary source of information (Ropret and Aristovnik 2019; Okuyucu and Yavuz 2020; Sharma et al. 2020).

It should be mentioned, however, that the use of WoS comes with some limitations. First, there is a rapidly developing set of participatory budgeting publications focused on the features of different vote-calculating algorithms for participatory budgeting that are largely centered on computer science in the fields of computational social choice theory or artificial intelligence (i.e., Rey and Maly 2023; Fairstein et al. 2023). With the faster development of algorithmic governance (Milosavljević et al. 2023), these streams of research should receive greater scholarly attention. Second, WoS is only one out of many useful databases that allow systematic analyses of scholarly papers (Milosavljević et al. 2023).

Using the combination of two keywords that are separated by the Boolean "OR" operator ("participatory budgeting" OR "participatory budget") we identified all possibly relevant papers in WoS database within the PB research topic. The string of keywords was applied in WoS research engine to the publication topic, which includes title, abstract, author keywords, and keywords plus. In addition, the initial search was restricted according to the publication type (Article OR Proceeding Paper OR Early Access OR Review Article) and publication language (we searched exclusively for publications in English language). The explained search inquiry resulted in a total of 450 publications within a time span from 1998 to January 2023.

The second phase was dedicated to the detailed content analysis of the extracted publications aiming to refine the research sample to the publications that are strictly dealing with PB. This was done by the manual inspection of retrieved papers (we conducted the full-text analysis). After the exclusion of the irrelevant publications for the research topic, the final research sample included 396 papers. For the final set of publications, we downloaded from WoS the full record information such as authors, title, abstract, document type, keywords, WoS categories, research area, publisher, etc., as Excel and tab-delimited files.

The third phase incorporated bibliometric analysis, which was supported by VosViewer software. The bibliometric analysis was performed in four interrelated steps: (i) descriptive analysis of the research sample to obtain insights on publication dynamics over time and research sample structure according to types of publications; (ii) descriptive analysis of retrieved documents according to spatial distribution, cooperation between the researchers, and productivity of journals and authors; (iii) descriptive analysis of the most cited publications; and (iv) thematic or content analysis of the main topics and subtopics emerging from the existing literature. The flow chart that outlines those phases of bibliometric analysis is shown in Figure 1.

The methodology explained above provides a solid basis for the comprehensive bibliometric analysis and graphical presentation of the literature evolution within the PB research field. Additionally, the research results are used to shed light on the status of the PB research field and to provide valuable recommendations for further research.

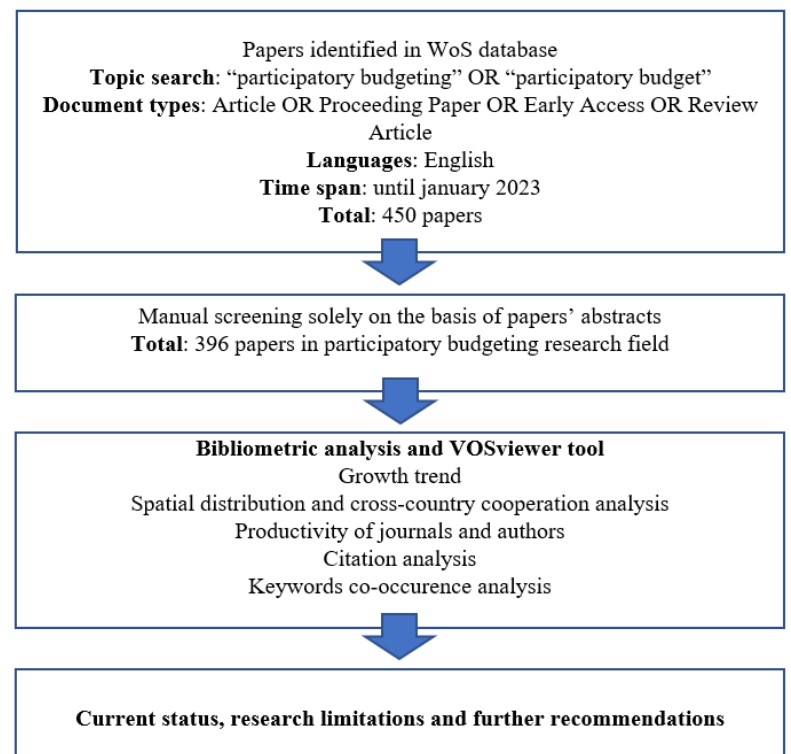

**Figure 1.** The flowchart of bibliometric analysis in PB research.

## 3. Results

In this section, we present the answers to the research questions set in this study. We addressed: (i) temporal and structural dynamics, (ii) spatial distribution of publications, (iii) journal and author productivity, and (iv) the main sub-topics.

### 3.1. Temporal and Structural Dynamics of the Participatory Budgeting Research

We first examined the output of publications over time. As shown in Figure 2, the first publication appeared in the WoS database in 1998. The number of publications has steadily grown until 2020, which is in line with the increasing number of PB cases after 2001 (Röcke 2014). The COVID-19 crisis, which ended PB processes in many cities or shifted the process online and significantly impacted the quality of citizen participation (Badia 2021) and instantly reduced research interest in this area, may be the cause of the decline in scientific output beginning in 2020.

The other indication of the dynamics of publications is the structure of publications on a particular topic. As presented in Table 1, the landscape of PB research is dominated by articles as a publication class with almost three-quarters of all publications (72.6%). Since the ratio of articles over proceeding papers is high (257/73 = 3.52), we can see that the field is highly saturated. It should be noted that different disciplines give different weights to conference papers and journal articles, and that this interpretation of the results might not be universally accepted. This metric is merely based on the logic that conferences are generally used to present some concepts and techniques that are in the development process, whereas journals usually publish concepts and techniques that have already been validated.

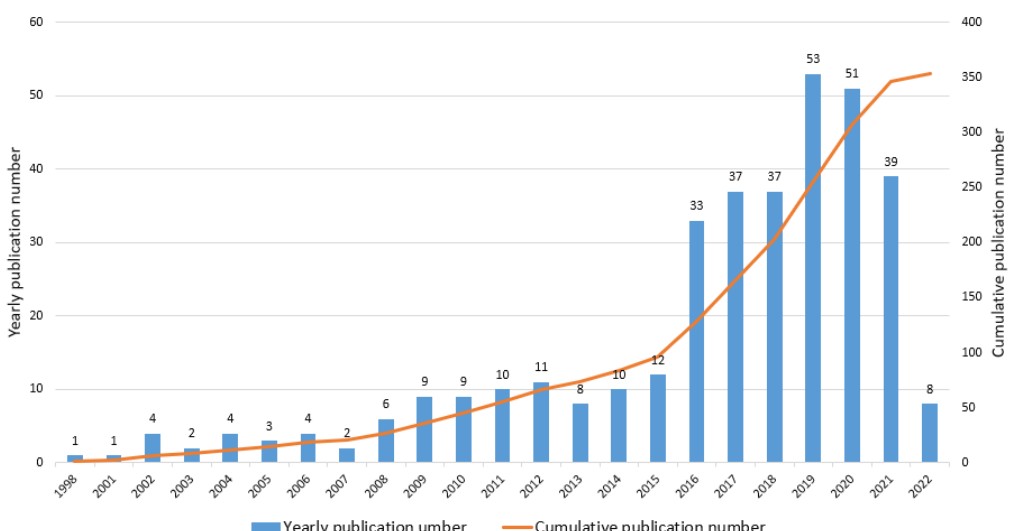

**Figure 2.** Distribution of publications on participatory budgeting.

**Table 1.** Structure of publications on participatory budgeting.

| No. | Document Type | Number of Documents | Proportion |
|-----|---------------|---------------------|------------|
| 1 | Articles | 257 | 72.60% |
| 2 | Proceedings Papers | 73 | 20.62% |
| 3 | Book Reviews | 15 | 4.24% |
| 4 | Review Articles | 9 | 2.54% |
| 5 | Total | 354 | 100.00% |

### 3.2. Spatial Distribution and Cross-Country Cooperation in the PB Realm of Research

When it comes to the spatial distribution of publications, nearly a third of all publications come from the USA (see Table 2). The spatial distribution refers to the research setting of the paper as indicated in the WoS database. As a result, the most productive author in the field is (Brian) Wampler from Boise State University, USA, with 14 documents, followed by (Dorota) Bednarska-Olejniczak from the Wroclaw University of Economics, Poland, with 8 documents in the research sample. The most influential papers from the USA usually deal with the role and contributions of PB for improving democracy and citizens' well-being through the analysis of the origins, global travel, and adoption of evolving PB practices (Wampler and Avritzer 2004; Touchton and Wampler 2013; Baiocchi and Ganuza 2014).

**Table 2.** Geographical distribution analysis.

| No | Country | Number of Papers | % of Total |
|----|---------|------------------|------------|
| 1 | USA | 117 | 33.05% |
| 2 | United Kingdom | 34 | 9.60% |
| 3 | Poland | 29 | 8.19% |
| 4 | Spain | 25 | 7.06% |
| 5 | Brazil | 23 | 6.50% |
| 6 | Canada | 21 | 5.93% |
| 7 | Germany | 17 | 4.80% |
| 8 | Australia | 16 | 4.52% |
| 9 | China | 15 | 4.24% |
| 10 | Italy | 14 | 3.95% |
| 11 | Others | 43 | 12.15% |
| | Total | 354 | 100% |

Note(s): Criterion for the inclusion: Top 10 countries.

Interestingly, the cradle of participatory budgeting—Brazil—only accounts for 6.5% of publications. The reasons are related to severe political, institutional, economic, and social crises in the country that marginalized the importance and benefits of PB and compromised the main pillars of the process (Dias and Júlio 2018). PB in Porto Alegre could not survive in an unfavorable country context and it was suspended by the decision of the municipality in 2017 (Núñez 2018).

Even though most papers are written by researchers from North American countries, PB is the most intensive and successful in Europe (De Vries et al. 2021). Based on the analysis of PB in France, Germany, and United Kingdom, Baiocchi and Ganuza (2014) claims that five main factors are essential for the introduction, long-term survival, and success of PB: (i) the existence of clear political support for this process and political willingness to go beyond the usual practices of citizen participation, (ii) the existence of strong power position whose authority is sufficient to implement PB, (iii) strong administrative support, (iv) wider political support, and (v) continuous financial support. The best example is Poland where PB was introduced in 2011 in a small city of Sopot with 33,000 inhabitants. The success of this project became a strong impulse for other cities to follow the model, making Poland the leader in Europe in terms of the number of PB cases. Poland is also interesting since it adopted the law in 2018 that made PB compulsory in 66 cities (with the status of a district city) and optional for the rest of the country (Kozłowski and Bernaciak 2021). A similar approach may be found in Peru, which accounts for two-thirds of PB cases in South America (De Vries et al. 2021).

It should be noted that the spatial distribution of papers does not reflect the geographical distribution of participatory budgeting cases. Some countries (i.e., USA, UK, Canada, and Australia) and their academics have better access to high-impact journals and Web-of-Knowledge indexed conferences, let alone English-language publications. On the other side, academic institutions from these countries often hire faculty from around the world who would take their international research interests with them.

When it comes to the cooperation between and among the researchers from various countries, we can clearly see the dominance of cooperation in the North-Atlantic region (see Figure 3). The strength of the relationship is somewhat weak, compared to other fields of research, which makes PB a relatively locally interesting topic. Thus, it does not come as a surprise when PB is addressed as a traveling innovation.

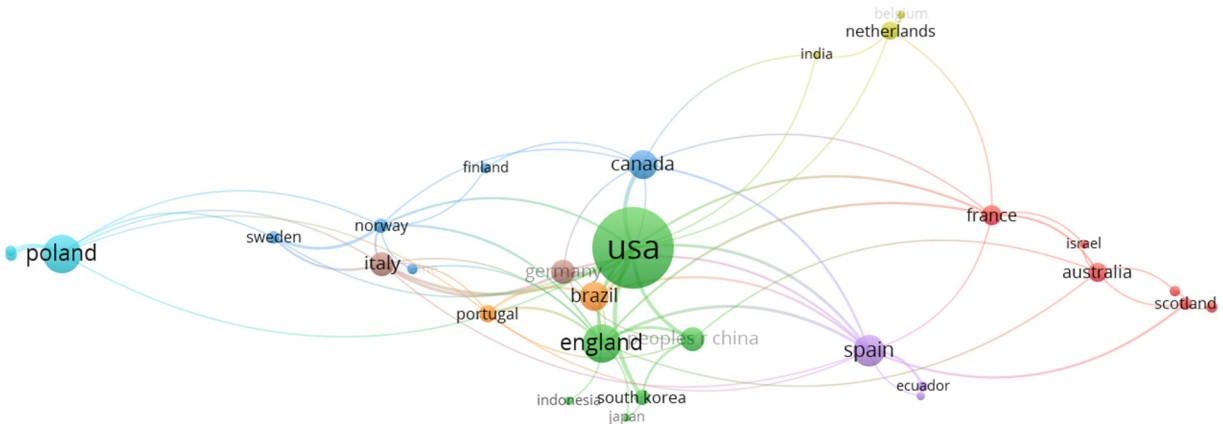

**Figure 3.** Cooperation network among countries/territories in PB research (n > 3).

### 3.3. Productivity of Journals and Authors

When it comes to the analysis of the productivity of research, we first tested the sample in a quantitative manner. There are no journals that could specifically be isolated as PB "heavens". Only seven journals indexed in the WoS database have published five or more articles related to PB. Most of them are in the field of "Urban Research", "Environmental Studies", or "Public Administration", as presented in Figure 4.

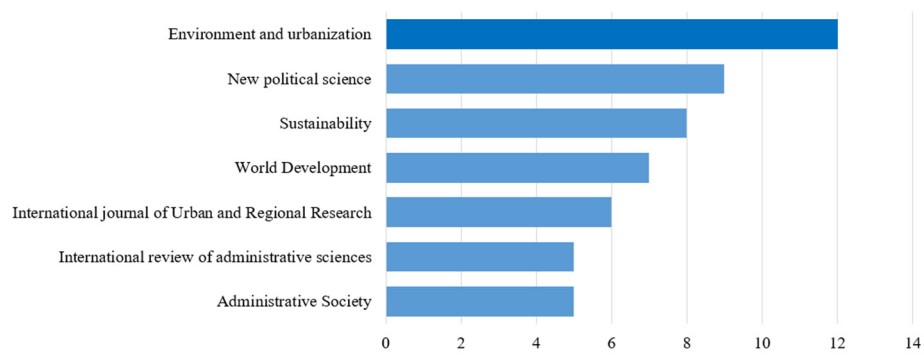

**Figure 4.** Articles published in journals (No of papers $\geq$ 5).

However, when compared against the measure of quality, we can clearly see the discrepancy. As an indication of the quality of the research, we used the total number of citations per publication. This indicator might be a subject for discussion. However, it clearly shows that PB has made the most valuable contribution to political and economic, rather than environmental or urban, science (see Table 3).

**Table 3.** An overview of the most cited publications.

| No | Authors | Title | Journal | Total Citations |
|----|---------|-------|---------|-----------------|
| 1 | Bingham et al. (2005) | The new governance: Practices and processes for stakeholder and citizen participation in the work of government | *Public Administration Review* | *466* |
| 2 | De Sousa Santos (1998) | Participatory Budgeting in Porto Alegre: Toward a Redistributive Democracy | *Politics & Society* | *224* |
| 3 | Cabannes (2004) | Participatory budgeting: a significant contribution to participatory democracy | *Environment & Urbanization* | *221* |
| 4 | Sintomer et al. (2008) | Participatory budgeting in Europe: Potentials and challenges | *International Journal of Urban & Regional Research* | *188* |
| 5 | Michels (2011) | Innovations in democratic governance: how does citizen participation contribute to a better democracy? | *International Review of Administrative Sciences* | *121* |
| 6 | Baiocchi and Ganuza (2014) | Participatory Budgeting as if Emancipation Mattered | *Politics & Society* | *114* |
| 7 | Souza (2001) | Participatory budgeting in Brazilian cities: limits and possibilities in building democratic institutions | *Environment & Urbanization* | *109* |
| 8 | Wampler and Avritzer (2004) | Participatory publics—Civil society and new institutions in democratic Brazil | *Comparative Politics* | *104* |

Notes: Inclusion criterion: Number of citations > 100. Date of observation: 1 January 2023.

The most cited publications in this field are usually conceptual by nature and draw conclusions from a small set of PB case studies. For instance, Bingham et al. (2005) addressed the issue of the emergence of new governance processes (such as participatory budgeting) and infer that they are "a natural, evolutionary human response to complexity". De Sousa Santos (1998) delineates the development of PB in Porto Alegre, and analyzes the PB process along redistributive efficiency, accountability, and quality of representation, autonomy of participatory budget, etc. Cabannes (2004) analyzes 25 municipalities in Latin America and Europe by several dimensions—such as 'the level of funds being considered, the extent of control and mode of involvement of local citizens, the relationship with local government, the degree of institutionalization and the sustainability of the process'. Sintomer et al. (2008) propose a valuable categorization of six ideal types of PB. Souza

(2001) highlights the importance of inclusion in the PB process. Michels (2011) challenges the theoretical proposition of the positive effects of citizen participation. Baiocchi and Ganuza (2014) evaluate emancipation as one of the fundamental pillars of PB.

### 3.4. Main Sub-Topics in the Field of PB

Finally, we analyzed the main sub/topics within the PB research area. We conducted (1) a neutral keyword-driven analysis and (2) a manually driven assessment of the main field of research for each PB study in our sample.

As for the neutral keyword-driven analysis, we measured the co-occurrence of keywords as stated by authors of publications. The results are displayed in Figure 5.

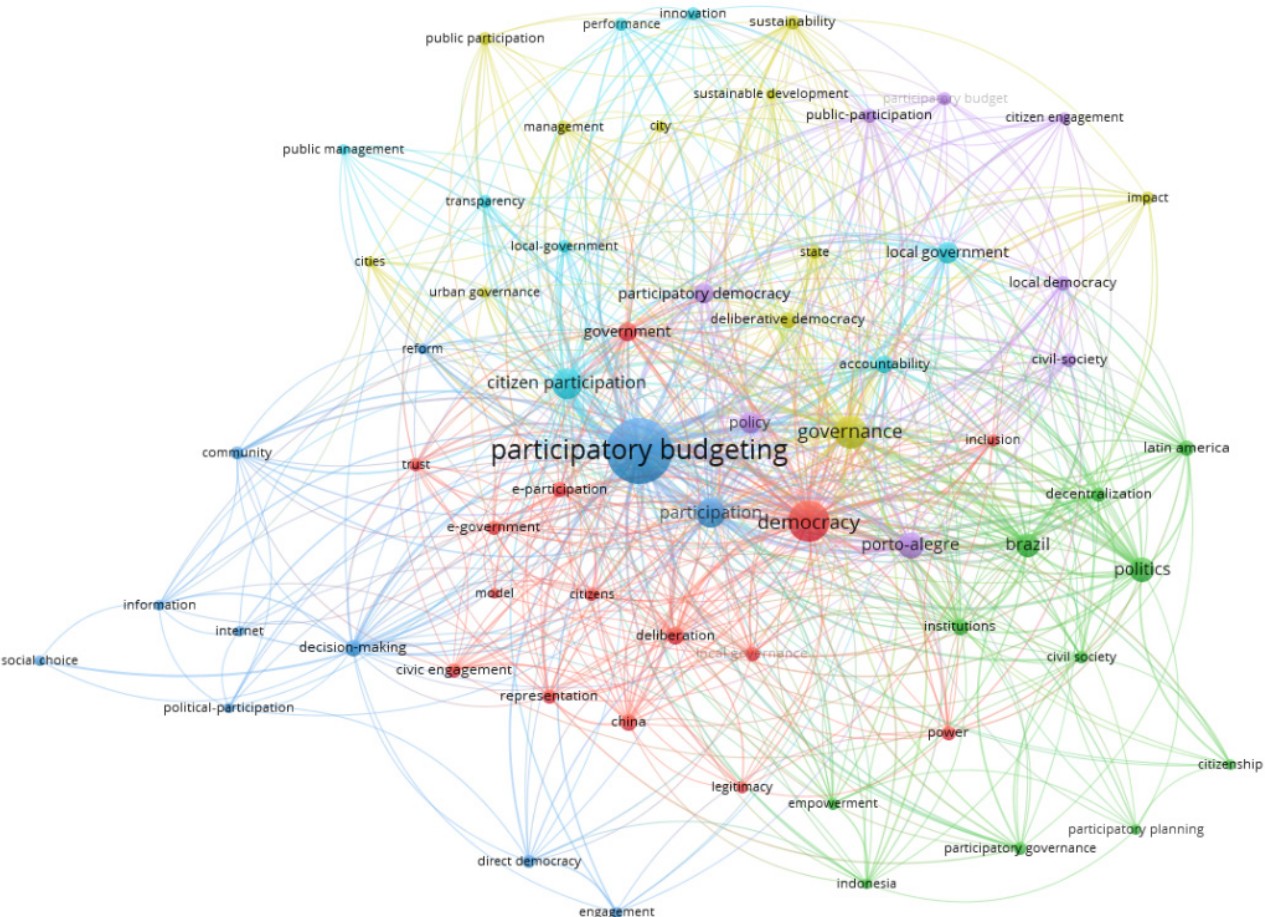

**Figure 5.** Keywords co-occurrence analysis of PB publications.

Based on the co-occurrence analysis, six distinct clusters were isolated. These clusters are given below (Table 4). Although some overlaps are evident, these clusters can be loosely delimited. Cluster 1 (red cluster) is a cluster of Democracy and Civic Engagement. Cluster 2 (green cluster) is a cluster of Decentralization and Institutions. Cluster 3 (dark blue cluster) is a cluster of Decision-Making. Cluster 4 (purple cluster) is a cluster of Urban and Sustainable Development. Cluster 5 (yellow cluster) is a cluster of Civil Society. Cluster 6 (light blue) is a cluster of Accountability, Public Management, and Performance.

The co-occurrence analysis to some extent reveals the main trends in PB research. From a grand scheme of things, the countries with democratic deficits usually provide publications related to empowerment, democracy, and citizen-centrism. When it comes to countries with a longer tradition in democratic political development, usual topics are centered around the decision-making process, engagement, and the use of technologies. It

should be noted that co-occurrence analysis does not provide clear distinctions between the main subtopics.

**Table 4.** Explanation of the main clusters in the co-occurrence analysis.

| Cluster | Keywords |
|---|---|
| Cluster 1 (red) | Citizens, civic engagement, deliberation, democracy, e-government, e-participation, government, inclusion, legitimacy, local governance, model, power, representation, trust |
| Cluster 2 (green) | Citizenship, civil society, decentralization, empowerment, institutions, participatory governance, participatory planning, politics |
| Cluster 3 (dark blue) | Community, decision-making, direct democracy, engagement, information, internet, participation, participatory budgeting, political-participation, reform, social choice |
| Cluster 4 (purple) | Cities, city, deliberative democracy, governance, impact, management, public participation, state, sustainability, sustainable development, urban governance |
| Cluster 5 (yellow) | Citizen engagement, civil-society, local democracy, participatory budget, participatory democracy, policy, Porto-Alegre, public-participation |
| Cluster 6 (light blue) | Accountability, citizen participation, innovation, local government, local-government, performance, public management, transparency |

As for the manually driven assessment of the main field of research for each PB study in our sample, the distribution is given in Figure 6.

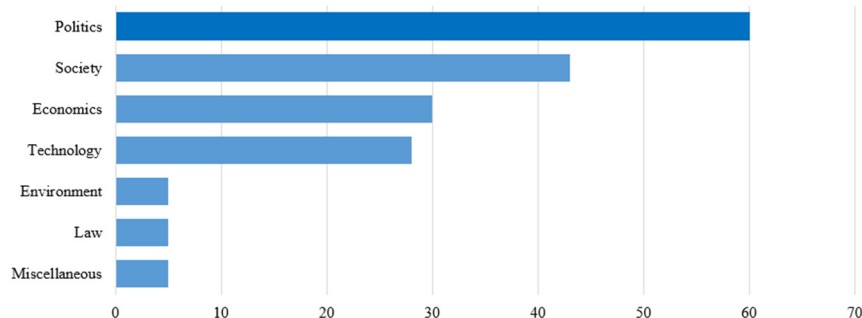

**Figure 6.** Distribution of publications by the discipline.

As displayed in Figure 6, studies on PB usually cover the field of politics, and issues such as the development of democratic institutions and the redefined role of government (Milosavljević et al. 2020), and governmental efficiency (Jung 2021). However, a number of critics have also been addressed within this cohort, particularly the papers discussing the unsuccessful "export" of this innovation via large global institutions (Milosavljević et al. 2021; Murray Svidroňová et al. 2023a).

From the societal perspective, participatory budgeting is viewed as a creative way to advance inclusive democracy, further modernization, and more accountability in the public sector by experts in the field of budgeting. Via "co-decision" processes, participatory budgeting is thought to enable the public sector and civil society to collaboratively decide on spending priorities. Conflicts are predicted to be lessened and budgetary decisions are predicted to be more widely accepted with cooperation. The most prominent ideas in our sample are tackling broader inclusivity (McNulty 2015), the potential amelioration of institutionalized inequality, inequity, and injustice (Callaghan and Horne 2023), power

dynamics, and political influence, and participation as a mean and higher institution of learning (Kasozi-Mulindwa 2016).

The studies covering the economic perspective often outline potential disadvantages of participatory budgeting. When residents are given the authority to decide how money is spent, they could give greater priority to local projects than to those that benefit a wider community. Also, studies find that there is no empirical evidence that PB increases the well-being of a community (Boulding and Wampler 2010) The possibility of corruption or bias is another problem. There is a possibility that specific groups or people could sway the decision-making process to advance themselves or their interests if the participatory budgeting process is not open and accountable. On the one hand, some studies even find that the revealed corruption increases the chance for a local government to implement PB (Timmons and Garfias 2015). On the other hand, several studies have proposed approaches for the improvement in accountability of the PB processes (Russo 2014).

As for the technology, we examined either as (1) the infrastructure that enables the digitalization of PB, or as (2) an advanced voting procedure. In terms of the infrastructure, the most important issue addressed in the concurrent body of knowledge is the efficiency of the online platforms for participatory budgeting. Studies advocate that digital platforms "must be revisable and reviewable while supporting accountability among participants and visibility of proposals and activities" (Menendez-Blanco and Bjørn 2022). Sampled papers also provide an overview of ICT used in PB (Sousa et al. 2019), as well as the effects of that use on eGovernance (Mærøe et al. 2020). As for the analysis of the voting procedure, papers usually address the typology of the voting used in the concurrent cases of PB or provide overviews, guidance, and novel methodologies and approaches for improved voting procedures (Kovacevic et al. 2020; Benade et al. 2021).

Finally, when it comes to the document elaborating on a legal point of view of PB, a general conclusion is that many countries have implemented participatory budgeting at the local level, with cities and municipalities adopting the process to involve citizens in decisions about local budget allocations. However, the legal framework for participatory budgeting can vary depending on the country. In some cases, the legal framework is based on a quasi-referendum (Sześciło and Wilk 2018), or a set of quasi-legislative and quasi-judicial new governance processes in international, federal, state, and local public institutions (Bingham et al. 2005).

## 4. Discussion

The practice of participatory budgeting has been around for more than 50 years, and there are thousands of case studies from around the globe (Oh et al. 2019; Buele et al. 2020). In this work, we analyzed the PB papers in a bibliometric manner. Research in PB has reached a point of saturation. Prior to the start of the COVID-19-induced pandemic, it attracted the greatest scientific attention. However, the knowledge base on PB has never grown in a linear manner. In fact, we have witnessed several waves of knowledge expansion in the field. The first peak was reached in 2010 driven by normative explanations and followed by pre-pandemic growth in empirical evidence on PB (Bhattarai et al. 2023). Its gradual decline is expected in the following years, which has already been noted in other studies (i.e., Wampler and Goldfrank 2022).

Naturally, the largest share of publications and evidence comes from the largest countries, since these countries have the most cases of participatory budgeting. Brazil remains one of the countries with the highest number of participatory budgeting initiatives, with hundreds of municipalities and state governments having implemented PB in some form or another. In addition, participatory budgeting has gained significant attraction in the United States in recent years, with several dozen participatory budgeting processes having been implemented in cities and towns across the country (Godwin 2018). PB is a multidisciplinary field, covering a broad range of topics.

Naturally, the primary focus of participatory budgeting studies is on political science and public administration. However, the interdisciplinary nature of the subject means that

researchers from many different fields can contribute to the study of participatory budgeting. A concurrent body of knowledge has mostly been focused on the political, societal, and economic aspects of PB. However, some scholarly fields, such as law, technology, urban planning, public health, and environmental science are insufficiently exploited topics. The most iconic publications in the field are usually conceptual by nature and try to explain the nature, benefits, or pitfalls of deliberative democracy.

The findings of our study contribute to the further development of scholarly knowledge in the field of participatory budgeting. PB is still an active field of research even three decades after the inception of the idea. This concept has brought about a number of tools for direct democracy and increased transparency of public financial decision-making processes (Milosavljević et al. 2017; Brun-Martos and Lapsley 2020). However, the PB field of research reached its peak point in 2019 and has ever since experienced a slowing downward process. It might only be a speculative judgment to claim that COVID-19 has shifted public administration attention from deliberative democracy to new challenges and issues. Other studies see this as an opportunity, since PB empowers citizens to play significant roles in emergency times (Anessi-Pessina et al. 2020). This study can be useful to a number of stakeholders. However, it has the most meaningful implications for researchers and lecturers in the field of public administration, budgeting, law, environmental science, and technology.

This study levels a scholarly terrain for future studies on PB. However, the bibliometric approach used in this study brings about several potential flaws. This study is limited by a very broad research phrase used to generate papers on PB. A possible avenue for future works is the inclusion of new and the extension of existing research phrases used to generate specific studies on PB. The literature search was conducted carefully. However, the search terms and the use of only English-language papers might have led to the exclusion of some relevant publications. Finally, we searched only the WoS database. Expansion to other databases such as Scopus, Google Scholar, CrossRef, or others might advance this study from being purely bibliometric to being semantic or systematic.

## 5. Conclusions

Participatory budgeting is a process in which community members decide in a rather direct manner how to allocate public funds. For more than three decades it has been titled as 'a democratic innovation' that promotes citizen empowerment, increases transparency, drives equity and innovativeness, and enables civic education. This study confirmed that the field of PB research has reached the point of saturation. In general terms, PB should become a standard topic, rather than a novelty for scholarly literature. Reaching the point of saturation does not imply any decrease in the number of papers published on this topic. It only means that the field of PB is going to require further specialization, which will make PB an advanced tool for direct democracy implementation.

**Author Contributions:** Validation, M.M.; formal analysis, Ž.S.; investigation, M.M.; resources, M.M.; writing—original draft preparation, M.M. and J.K.; visualization, Ž.S. All authors have read and agreed to the published version of the manuscript.

**Funding:** This paper was supported by the University of Belgrade—Faculty of Organizational Sciences.

**Informed Consent Statement:** Not applicable.

**Data Availability Statement:** The dataset used in this paper is available upon request to the corresponding author.

**Acknowledgments:** We appreciate the comments and suggestions given by the Editor and anonymous reviewers.

**Conflicts of Interest:** The authors declare no conflict of interest.

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
