# Peer review of "Bibliometric Review of Participatory Budgeting: Current Status and Future Research Agenda"

_ijfs, doi:10.3390/ijfs11030104_

Round 1
Reviewer 1 Report
The paper deals with a very current topic of participatory budgeting in a novel way by using bibliometric analysis. It presentes a very solid research, only minor revisions are required, listed in chronological order:
p. 4, 135 - "As shown in Figure 2, the first publication appeared in the WoS database in 1996" - this should be 1998, surely.
p. 5, 182 - "Poland is also interesting since it adopted the law in 2018 that made PB compulsory in 66 cities and optional for the rest of the country (Kozłowski and Bernaciak, 2021)" - one might ask, why in these 66 cities, maybe just add "66 cities with the status of a district city"
p.10, 318 - "In addition, participatory budgeting has gained significant attraction in the United States in recent years, with over 1,500 participatory budgeting processes having been implemented in cities and towns across the country" - where did you get the number from? This should be quoted, it needs a source of data.
p. 11, - 364 "All these advancements will utterly lead to the migration of PB from local to central levels of government." - I am not sure that this is well supported by the analysis and findings. Either elaborate this idea but more or omit this statetment.
And just a thought for consideration, you state "Since the ratio of articles over proceeding papers is high (257/73=3.52), we can see that the field in highly saturated." How this can be influenced by the fact that in the most part of CEE countries the academics are "pushed" to published in the high-ranking journals and these have priority over the confence proceedings?
p. 6, 196 - "When of comes" - when it comes
p. 10, 310 - "the first peak was reached in 2010" - capital T at the beginning of the sentence.
p. 10, 314 "The largest share of publications and evidence comes from the largest countries, which has been expected. This is somewhat expected since these countries have the most cases of participatory budgeting" - rephrase these two sentences, merge into one.
p. 11, 361 - "the down-top decision" - did you mean top-down or bottom-up? Down-top does not exist.
Author Response
Dear Reviewer,
Thank you for providing us with the opportunity to adjust and refine our manuscript. We appreciate the time and effort that you dedicated in the review process.
All your insightful comments and useful suggestions have been addressed in our revised version. The changes related to your comments are highlighted in green in the manuscript.
Best regards!

Reviewer 2 Report
Acknowledge that there are gaps that come from Web of Science as a starting point for the bibliometric review. Notably, there is a rapidly developing set of participatory budgeting publications focused on the features of different vote-calculating algorithms for participatory budgeting that are largely centred in computer science (computational social choice theory; artificial intelligence) that are being missed. The paper does not need to re-run the bibliometric review to capture these, just to acknowledge the gaps in data set.
Because of the above, it is worth tempering some of your statements. Ex. “Empirical studies still dominate the spectrum of PB research”(p 2, line 48) isn’t really true of the computational social choice work on PB which tends to work on axiomatic mathematical proofs and simulation models. Empirical studies dominate this sample of PB research” is a more accurate statement.
Research questions are clear and well-structured.
Gap in the literature can be made clearer. On Page 2, lines 48-72 there is a good review of similar work but it is still unclear what this paper’s novel contribution is. Given the variety of bibliometric tools available to conduct an analysis of this type, a bit more clarity can be put into explain what the specific gap is and what this particular approach to bibliometrics is offering. Since you acknowledge that others have used the same data set (albeit slightly older) it is worth explaining how your approach to using this data set is different than what has appeared before and to also argue that your research questions haven’t been addressed in previous work.
Methodology section is clearly written and easy to follow.
Page 4 lines 145-148 – I am not sure if I agree with this statement. It may be true, but it is important to note that different disciplines weight the importance of proceedings vs articles quite differently.
Page 5 – Table 2. More clarity needed on the composition of the table here – is it the location of the first author which is used here? Given the N=354 it is unlikely that all author locations are counted in multi-authored articles, so be explicit.
Additionally, with Table 2 I would be wary of drawing too strong conclusions about geography when the *institution* is the focus, not the *author*. The USA and UK (NOT England – I am assuming these are being used interchangeably but England wouldn’t contain, for example, Scottish institutions which is particularly important because Scotland has done relatively more PB experimentation than England) are unsurprising as the top 2, though both countries (along with Canada in 6th spot and Australia in 8th spot) are laggards internationally when it comes to actually implementing PB. However, those countries have well-funded research university systems, often hire faculty from around the world who would take their international research interests with them, and are built around producing English-language publications captured in WoS. That said, the explanation of Poland as #3 is particularly interesting since it would seem to be a particular outlier without the history of a country like Brazil nor with the same publication advantages as US/UK/Canadian/Australian institutions.
Page 8 – findings from Table 4 could use more explanation. It is unclear what each of these clusters is telling us. For example, it is unclear what trend is captured by “China, citizens, civic engagement, democracy, e-government, e-participation, government, inclusion, legitimacy, local governance, model, power, representation, trust” as this seems like a pretty disparate range of keywords with a lot of overlap with other clusters. What makes these clusters distinct (if anything)?
Page 9, line 284: Keep an eye to language. “advance voting” has a specific meaning in much of the Anglo-sphere, which is voting during elections that occur before election day itself.
Page 11, lines 358-365: It is unclear that these findings come form the bibliometric work that has been conducted or whether they are the authors’ informed speculation. Either clarify the findings that led to these conclusions, or write these as areas for future research that appear to be gaps in the existing literature worth future exploration.
Overall additional English-language copy-editing from someone familiar with PB is needed. The writing is of good quality generally, but there are pieces of terminology and grammar that occasionally distract or confuse the messaging.
Author Response
Dear Reviewer,
Thank you for providing us with the opportunity to adjust and refine our manuscript. We appreciate the time and effort that you dedicated in the review process.
All your insightful comments and useful suggestions have been addressed in our revised version. The changes related to your comments are highlighted in light-blue in the manuscript.
Best regards!

Round 2
Reviewer 1 Report
The paper has been improved as requested in previous revision. It can be published as it is.
Reviewer 2 Report
Paper has been improved from its earlier form and is acceptable for publication.